# Trends in the prevalence and treatment of depressive symptoms in Peru: a population-based study

David Villarreal-Zegarra [1,2] Milagros Cabrera-Alva,[1]
Rodrigo M Carrillo-Larco [2,3] Antonio Bernabe-Ortiz [2,4]

¹Instituto Peruano de Orientación Psicológica, Lima, Peru
²CRONICAS Center of Excellence in Chronic Diseases, Universidad Peruana Cayetano Heredia, Lima, Peru
³Department of Epidemiology and Biostatistics, Imperial College London School of Public Health, London, UK
⁴Universidad Científica del Sur, Miraflores, Peru

**Correspondence to**
David Villarreal-Zegarra;
davidvillarreal@ipops.pe

## ABSTRACT

**Objectives** This study aimed to estimate the trends in the prevalence and treatment of depressive symptoms using nationally representative surveys in Peru from 2014 to 2018.

**Design** A secondary analysis was conducted using five nationally representative surveys carried out consecutively in the years between 2014 and 2018.

**Setting** The study was conducted in Peru.

**Participants** Individuals, men and women, aged ≥15 years who participated in the selected surveys. Sampling was probabilistic using a two-stage approach.

**Main outcome measures** Two versions of the Patient Health Questionnaire (PHQ-9) that focused on the presence of depressive symptoms were administered (one in the last 2 weeks and other in the last year). Scores ≥15 were used as the cut-off point in both versions of the PHQ-9 to define the presence of depressive symptoms. Also, the treatment rate was based on the proportion of individuals who had experienced depressive symptoms in the last year and who had self-reported having received specific treatment for these symptoms. The age-standardised prevalence was estimated.

**Results** A total of 161 061 participants were included. There was no evidence of a change in age-standardised prevalence rates of depressive symptoms at the 2 weeks prior to the point of data collection (2.6% in 2014 to 2.3% in 2018), or in the last year (6.3% in 2014 to 6.2% in 2018). Furthermore, no change was found in the proportion of depressive cases treated in the last year (14.6% in 2014 to 14.4% in 2018). Rural areas and individuals with low-level of wealth had lower proportion of depressive cases treated.

**Conclusions** No changes in trends of rates of depressive symptoms or in the proportion of depressive cases treated were observed. This suggests the need to reduce the treatment gap considering social determinants associated with inequality in access to adequate therapy.

## INTRODUCTION

Depression is a significant public health issue that affects approximately 322 million people worldwide[1] and has tremendous social and economic cost implications.[2][3] Based on current projections, it is estimated that depression will be the leading cause of years lost to disability (disability-adjusted life year)

### Strengths and limitations of this study

► This analysis included information from five nationally representative surveys of the Peruvian population.
► Our findings were limited as screening for depression in individuals did not involve psychiatric evaluations or structured interviews. As a result, only depressive symptoms were used as indicators to identify cases.
► Only 5 years were evaluated and perhaps more time may be required to identify a significant trend.

by 2030.[4] Epidemiological surveillance of the trends in depressive symptoms is of high relevance to public health as changes in the prevalence and treatment can aid researchers and stakeholders make more informed decisions on how to address this challenge.

The prevalence of depressive symptoms can vary depending on several biological, sociodemographic or lifestyle factors. For example, the prevalence of depressive symptoms is reportedly higher in women,[5] older adults[6] and people of a low socioeconomic status.[7] On the other hand, the fact that a person with depressive symptoms can access treatment also depends on the accessibility to health services. A systematic review identified that lack of human resources, centralisation of the health system and integration in primary care are barriers to receiving appropriate treatment for depressive symptoms.[8]

Currently, there is no consensus on the trends of the prevalence of depressive symptoms. A meta-analysis of 116 epidemiological investigations during the years 1990 and 2010 did not reveal changes in the prevalence of major depressive disorder.[9] Nevertheless, when assessing trends in previous years,[10][11] the Global Burden of Disease Studies found a decrease of 4.9% in the age-standardised rate of depressive

disorders from 2006 to 2016. In contrast, other longitudinal studies showed an increase in the prevalence of major depressive disorder.[12 13] These mixed results can be attributed to inter-country variations of several demographic and socioeconomic factors, as well as the different criteria used to define depression in research studies. Due to the lack of global consensus, country-specific estimates and trends are significant and can inform local policies and guidelines.

In Peru, the prevalence of depressive symptoms ranges from 14% in urban areas to 12.5% in rural areas.[14] The prevalence of major depressive disorders reported in the last year was 2.7%, of which a third receive minimally adequate treatment.[15] Moreover, access to treatment for those suffering from depressive disorders is very limited in low-income and middle-income countries, with only 1 in every 27 patients with major depressive disorder managing to receive treatment.[15] Since 2015, Peru has been implementing mental health reforms that focus on primary care and the redistribution of its resources from hospital care to community care centres.[16]

It is therefore fundamental to have up-to-date metrics on the prevalence and level of treatment coverage of depressive symptoms in order to determine an appropriate response to this disease. This study aims to estimate trends in prevalence and treatment of depressive symptoms using nationally representative surveys in Peru between 2014 and 2018.

## METHODS
### Study design
Data from the National Demographic and Health Survey of Peru (ENDES—Spanish acronym), which collects information on several variables, including poverty, fertility, violence and health, were utilised in this study. Implementation of ENDES started in 1996 and was originally carried out every 4 years, with the second being conducted in 2000, and the third in 2004. Since 2013, ENDES began to incorporate questions regarding mental health into its evaluation. However, it initially only used a very small subsample that was not representative of the general population. From 2014 onwards, the data collected on mental health was nationally and regionally representative. Our study therefore decided to use data collected by ENDES between the years 2014 and 2018.

The data collection from each of the years included in the study used a face-to-face approach and started in February or March, with completion ending in December.[17 18] Data collection was face-to-face, with each participant evaluated being interviewed.

The participants from whom the data were being collected from in the annual ENDES evaluations were different each year. Therefore, the data being evaluated in this study came from five different groups (one from each year of evaluation). So, while there was a chance that the same participant would participate in two different years or that two or more participants from the same

household would be evaluated in 1 year, this probability was negligible.

### Participants
The sampling used was probabilistic in two stages and representative at both national and regional levels. The sampling frame in the first stage was the selection of primary sampling units (clusters) based on information from the last census conducted in Peru. In the second stage, the selection of secondary sampling units (households) was carried out based on the information from cartographic updates and the register of buildings and households made previously.[17]

In rural areas, the primary sampling units were comprised of groups ranging from 500 to 2000 people, whereas the secondary units included households. Contrastingly, the primary sampling units in urban areas were blocks or groups of blocks involving more than 2000 individuals with an average of 140 households. Secondary sampling units were the same as in rural settings.[17] Details on the sampling process can be found in the technical documents of the ENDES.[18]

Information on the dates of birth of all household members and the order in which these data were collected was used for the selection of participants. Only one participant aged 15 years or older was selected from each household. The participant with the closest birthday to the evaluation date was selected in the ENDES. In the event of a tie between two or more participants' birthdays (ie, same birthday), the participant whose data were collected first was selected.[19]

Participants of both sexes, who were of more than 15 years of age, living in both rural and urban areas and from every region in Peru, were included in ENDES. This study only included participants with complete data on the Patient Health Questionnaire (PHQ-9) and the sociodemographic variables of interest (sex, age, area, economic level, region and year of evaluation).

The number of participants evaluated by the ENDES in each year is about 30 000 (see table 1).

### Variables
#### Depressive symptomatology in the last 2 weeks
Depressive symptomatology was evaluated using the PHQ-9, available in the ENDES 2014–2018. The PHQ-9 was adapted for the population in Peru and had shown optimal values of reliability and validity when tested.[20] This tool has nine items based on the Diagnostic and Statistical Manual of Mental Disorders, fourth edition criteria and qualifies the response options from 0 to 3. PHQ-9 can, therefore, present a minimum score of 0 and a maximum score of 27. Depending on the total score, the PHQ-9 results may be without indicators of depressive symptoms (scores 0–4); mild depressive symptoms (scores 5–9); moderate depressive symptoms (scores 10–14); moderate-to-severe depressive symptoms (scores 15–19); and severe depressive symptoms (score 20 and above).[21]

**Table 1** Descriptive characteristics of the participants included in the study by year

| | 2014 (n=27 633) | | 2015 (n=33 573) | | 2016 (n=32 373) | | 2017 n=33 794) | | 2018 (n=34 476) | |
|---|---|---|---|---|---|---|---|---|---|---|
| | n | % | n | % | n | % | n | % | n | % |
| **Sex** | | | | | | | | | | |
| Male | 12 806 | 46.3 | 14 788 | 48.9 | 14 126 | 48.9 | 14 424 | 48.5 | 14 696 | 48.4 |
| Female | 14 827 | 53.7 | 18 573 | 51.1 | 18 247 | 51.1 | 18 794 | 51.5 | 19 780 | 51.6 |
| **Age** | | | | | | | | | | |
| 15–34 | 11 022 | 43.1 | 16 576 | 43.1 | 15 498 | 44.4 | 15 913 | 43.6 | 16 121 | 42.7 |
| 35–54 | 9395 | 33.9 | 10 887 | 33.9 | 10 740 | 33.8 | 10 819 | 34.5 | 11 692 | 34.6 |
| 55–74 | 5460 | 18.0 | 4685 | 18.0 | 4791 | 16.8 | 5030 | 17.0 | 5332 | 17.7 |
| 75+ | 1756 | 5.0 | 1213 | 5.0 | 1344 | 5.0 | 1456 | 4.9 | 1331 | 5.0 |
| **Area** | | | | | | | | | | |
| Rural | 10 663 | 24.9 | 11 453 | 34.5 | 11 113 | 35.2 | 11 349 | 20.8 | 11 923 | 19.6 |
| Urban | 16 970 | 75.1 | 21 908 | 65.5 | 21 260 | 64.8 | 21 869 | 79.3 | 22 553 | 80.4 |
| **Wealth index** | | | | | | | | | | |
| Very low | 8151 | 18.7 | 9370 | 26.3 | 8967 | 26.4 | 10 045 | 18.5 | 11 019 | 18.7 |
| Low | 6782 | 19.2 | 8376 | 21.2 | 8464 | 22.0 | 8575 | 20.8 | 8514 | 20.6 |
| Middle | 5153 | 19.9 | 6459 | 18.3 | 6361 | 18.5 | 6368 | 21.0 | 6379 | 20.9 |
| High | 4103 | 21.0 | 5089 | 17.7 | 5056 | 17.4 | 4846 | 20.3 | 4909 | 20.4 |
| Very high | 3444 | 21.2 | 4067 | 16.4 | 3525 | 15.8 | 3384 | 19.4 | 3655 | 19.4 |

Two-stage sample design was taken into account for percentage estimations.

For this study, scores of ≥15 were used to define the presence of depressive symptoms, as this cut-off point allows for better values of sensitivity (68%) and specificity (95%).[21] In Peru, no studies have been conducted on the sensitivity and specificity of PHQ-9. Scores ≥15 were used to define the presence of depressive symptoms since it is an indicator of the presence of a major depressive episode that requires treatment.[21 22]

### Depressive symptomatology in the last year
A modified version of the PHQ-9 was used to assess depressive symptoms over the last year to help identify the occurrence of depressive symptoms over a more extended period of time. This version of the PHQ-9 was used in ENDES 2014–2018 to evaluate depressive symptoms experienced at some point in the last 12 months. Participants were asked to remember an event in the last 12 months in which they had discomfort or problems. Once the participants identified that troublesome or annoying event, the PHQ-9 was applied based on the 2 weeks around that event. The definition of depressive symptoms experienced some time in the last year was based on two criteria: (1) a score of ≥15 in the 2-week version of PHQ-9 or (2) a score of ≥15 in the modified version of PHQ-9 for the last year. The age-standardised prevalence was also analysed.

### Proportion of depressive cases treated in the last year
A participant was considered to receive treatment for symptoms of depression based on the following two criteria.

1. Depressive symptoms had been recorded in the previous year or the past 2 weeks (PHQ-9 ≥15)
2. They had self-reported receiving treatment for depression from a health professional in the last 12 months.

There were three response options ('yes', 'no' or 'I do not remember'). Participants were only considered having received treatment if an affirmative answer was provided. It should be noted that ENDES data did not include information about the frequency or type of treatment received.

### Other variables
In addition to the previous variables mentioned, a set of sociodemographic variables were also taken into consideration. The wealth level of participants was defined in quintiles (very low, low, middle, high and very high) based on a wealth index available in the ENDES.[23] This index was calculated by using the availability of goods and services, the housing characteristics that the participants reported having.[17] The index was built for each year and categorised into quintiles. The calculation of this index can be found in Rutstein and Johnson.[23] As a continuous variable, age was split into four groups (15–34, 35–54, 55–74 and 75+). The sex (male vs female), the study area (urban/rural) and the year of ENDES evaluation were also taken into consideration.

### Statistical methods
#### Main analysis
First, the number and proportion of participants excluded from the analyses were recorded (online supplementary

file 1). Second, a descriptive analysis of the participants was carried out for each year of the ENDES. Third, the age-standardised prevalence of depressive symptoms was estimated using the WHO population as the reference population.[24] The age-standardised prevalence was estimated since it enabled us to compare our results with studies conducted in other countries. A 95% CI was calculated for the prevalence at both regional and national levels. Then, an analysis was conducted on subgroup participants reporting depressive symptoms over the past year in order to determine the proportion of those who reported receiving treatment (ie, sex, area, age groups). Finally, the trend over time was age-standardised and then evaluated using the score test for trend; for this, the year 2014 was used as the reference category. The trend test compares the odds of cases in 1 year with the odds of cases in the next year. This test assumes that the trend is linear and can be used in STATA version 13.0 with the 'tabodds' command.[25]

### Subanalysis

Four subanalyses were conducted to complement the main results. In order to evaluate the measurement properties of the modified version of the PHQ-9 that collects information on depressive symptoms in the last year, a subanalysis was conducted (online supplementary file 2). A confirmatory factor analysis was performed to evaluate the validity of the modified version of the PHQ-9, considering the ordinal nature of the items and using the estimator of weighted least square means and variance adjusted.[26] These analyses evaluate whether the instrument fits the one-dimensional model proposed by the PHQ-9 as the one-dimensional model seems to be adequate,[20] when optimal values are reached in the different goodness-of-fit indices. The Comparative Fit Index (CFI) and the Tucker-Lewis Index were used; these indices must be greater than 0.90 in order to be considered of an adequate level.[27] The root mean square error of approximation (RMSEA) with a CI of 90% and the standardised root mean square residual were also used, both indices considered fair values as those lower than 0.08.[27] On the other hand, reliability was evaluated by the internal consistency coefficient of alpha and omega. Both coefficients consider that adequate levels of reliability are reached if they score higher than 0.80.[28]

An additional post hoc analysis was performed, which requires participants with depressive symptoms to meet the Diagnostic and Statistical Manual of Mental Disorders, fifth edition (DSM-5) criteria. This analysis aimed to identify whether using the clinical criteria proposed by DSM-5 can alter the results (higher or lower) or be equivalent to our main results, which use a score of ≥15 in the PHQ-9. For this subanalysis, participants must have feelings of sadness or anhedonia (have a score of 2 or more on items 1 and 2 of the PHQ-9, ie, 'more than half the days' and 'nearly every day') and at least five of the other seven indicators (have a score of at least 1 on 5 of the other 7 items of the PHQ-9).

In addition, a subanalysis was performed to compare sociodemographic characteristics in people receiving treatment for depressive symptoms in the last year. The Pearson $\chi^2$ test was used to make these comparisons. The generalised linear model assumed a Poisson distribution (crude and adjusted models). Assuming a Poisson distribution, link log and robust variance were used as suggested in the literature.[29] Prevalence ratios and 95% CI were reported.

Finally, the age-standardised prevalence of depressive symptoms in the last 2 weeks and in the last year, and the proportion of depressive cases treated for each region of Peru were evaluated (online supplementary file 3).

### Software used

All statistical analyses were performed using STATA V.13 (StataCorp, College Station, Texas, USA). The graphics were elaborated using the ggplot libraries in R (V.3. 5. 1) and QGIS V.2.18. All the analyses performed considered the design by a complex sampling of the ENDES, and the analyses performed were adjusted based on the weight factor provided by each ENDES assessment year. Adjustments were made with the STATA command 'svy' for all analyses, except for the factor analysis (subanalysis) where the 'lavaan' and 'lavaan.survey' package was used in R.

### Patient and public involvement

The patients or members of the public were not involved in the design, conduct, reporting or dissemination plans of our research.

## RESULTS

### Participants

The ENDES reports published between 2014 and 2018 included the data of 166 290 participants. The participation rate varied between 95.7% in 2014 and 97.4% in 2018 (average=96.8%). After excluding all records with incomplete information (n=5229, online supplementary file 1), a total of 161 061 participants were included. Of these, 51.7% were women, the mean age was 40.3 years (SD=17.1), and 73% lived in urban settings. The sociodemographic characteristics of the participants for each year are shown in table 1.

The measurement properties of the modified version of the PHQ-9 used to assess depressive symptoms during the last year were evaluated before the primary analyses. The modified version of the PHQ-9 was identified as having evidence of validity by confirmatory factor analysis (CFI>0.90; RMSEA<0.05) and evidence of reliability by internal consistency (ω and α>0.85). More information on the factorial analysis of the modified version of the PHQ-9 for year is presented in online supplementary file 2. These results support that the modified version of the PHQ-9 presents evidence of structural validity, supporting the one-dimensional model.

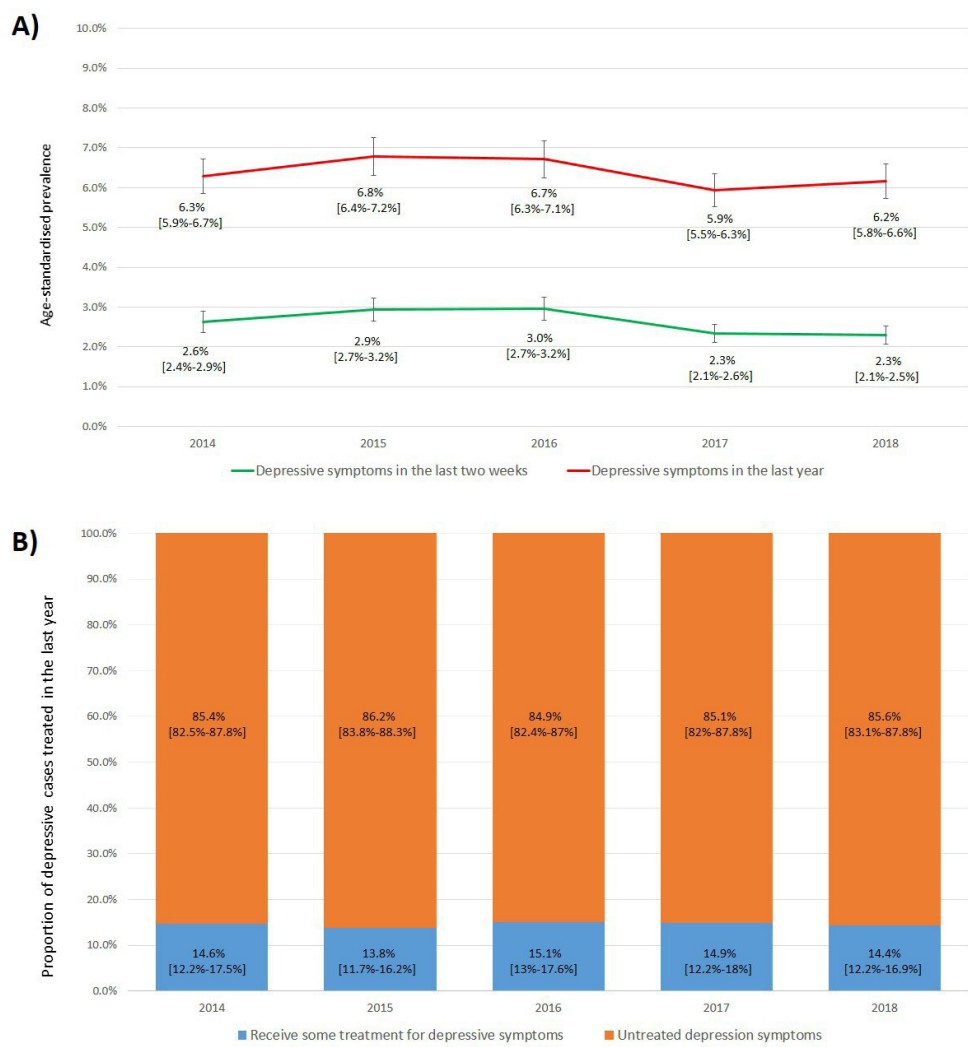

**Figure 1** Age-standardised prevalence of depressive symptoms and proportion of depressive cases treated in Peru between 2014 and 2018. (A) Age-standardised prevalence of depressive symptoms in the last year and the last 2 weeks in Peru by year. (B) Proportion of depressive cases treated in the last year in Peru by year. In all the analyses, the weighted proportion by complex sampling was used.

### Prevalence of depressive symptomatology

In Peru, between 2014 and 2018, the age-standardised prevalence of depressive symptoms in the last 2 weeks (score test for trend: p value=0.39, $\chi^2$=0.72) and in the last year (score test for trend: p value=0.38, $\chi^2$=0.74) did not show a significant trend. In 2018, the age-standardised prevalence in the last 2 weeks and in the last year were 2.3% (n=892, 95% CI: 2.1% to 2.5%) and 6.2% (n=2291, 95% CI: 5.8% to 6.6%), respectively (see figure 1A). On the other hand, in 2018, the unstandardised prevalence of depressive symptoms in the last 2 weeks was 2.7% (95% CI: 2.4% to 2.9%) and in the last year was 6.2% (95% CI: 5.8% to 6.6%), and therefore showed little to no difference from the age-standardised prevalence results.

Women had a higher age-standardised prevalence of depressive symptoms in the last 2 weeks and the last year in comparison to men. Adults over the age of 75 years and individuals in the lowest wealth quintile were those who exhibited the highest age-standardised prevalence in the last 2 weeks and the last year (see table 2). Figure 2A and B shows the age-standardised prevalence of depressive symptoms in the last 2 weeks and last year by region, respectively. There was no variation in the trends over time in any of the evaluated regions.

Trend results were not different when DSM-5 criteria were used. Thus, the age-standardised prevalence was very similar for both depressive symptoms in the last 2 weeks (3.4% in 2014 to 3.3% in 2018) and in the last year (6.8% in 2014 to 6.8% in 2018).

### Proportion of depressive cases treated

At the national level, the proportion of depressive cases treated in the last year did not show any significant changes over time (score test for trend: p value=0.19, $\chi^2$=1.66; see figure 1B). In 2018, 14.4% (n=292, 95% CI: 12.2% to 16.9%) of people with depressive symptoms in the last year reported having received some type of treatment from a healthcare professional. A higher proportion of women self-reporting for treatment was found when compared with men (only 2018, 15.9% in women

**Table 2** Age-standardised prevalence of depressive symptoms and proportion of depressive cases treated in the last year in Peru by selected sociodemographic characteristics (2014–2018, with cut-off point 15)

| | 2014 (%) | 2015 (%) | 2016 (%) | 2017 (%) | 2018 (%) |
|---|---|---|---|---|---|
| **Age-standardised prevalence of depressive symptoms in the last 2 weeks*** | | | | | |
| **Sex** | | | | | |
| Male | 1.5 | 1.9 | 1.9 | 1.3 | 1.1 |
| Female | 3.6 | 3.9 | 4.0 | 3.3 | 3.4 |
| **Age** | | | | | |
| 15–34 | 1.5 | 1.5 | 1.6 | 1.2 | 1.1 |
| 35–54 | 2.9 | 2.6 | 2.6 | 2.5 | 2.3 |
| 55–74 | 4.0 | 5.7 | 5.0 | 4.1 | 4.1 |
| 75+ | 6.7 | 9.0 | 11.2 | 6.0 | 6.5 |
| **Area** | | | | | |
| Rural | 2.9 | 3.5 | 3.4 | 3.2 | 3.3 |
| Urban | 2.5 | 2.5 | 2.7 | 2.1 | 2.0 |
| **Wealth index** | | | | | |
| Very low | 3.0 | 3.8 | 3.6 | 3.2 | 3.5 |
| Low | 3.1 | 3.3 | 2.8 | 3.1 | 2.7 |
| Middle | 3.3 | 3.2 | 3.5 | 2.8 | 2.4 |
| High | 2.5 | 2.1 | 2.6 | 1.6 | 1.7 |
| Very high | 1.6 | 1.7 | 1.7 | 1.2 | 1.2 |
| **Age-standardised prevalence of depressive symptoms in the last year*** | | | | | |
| **Sex** | | | | | |
| Male | 3.7 | 4.3 | 4.0 | 3.5 | 3.9 |
| Female | 8.5 | 9.1 | 9.3 | 8.2 | 8.2 |
| **Age** | | | | | |
| 15–34 | 4.5 | 4.4 | 4.6 | 4.1 | 4.3 |
| 35–54 | 6.9 | 6.7 | 6.5 | 6.0 | 6.1 |
| 55–74 | 8.1 | 10.6 | 9.7 | 9.3 | 9.0 |
| 75+ | 12.5 | 15.3 | 17.3 | 9.6 | 13.2 |
| **Area** | | | | | |
| Rural | 7.2 | 7.9 | 7.6 | 7.4 | 8.1 |
| Urban | 5.9 | 6.0 | 6.1 | 5.5 | 5.6 |
| **Wealth index** | | | | | |
| Very low | 7.4 | 8.1 | 7.8 | 7.3 | 8.4 |
| Low | 7.4 | 7.7 | 7.1 | 7.4 | 7.1 |
| Middle | 7.1 | 6.9 | 7.4 | 6.5 | 6.6 |
| High | 5.4 | 5.7 | 5.9 | 4.6 | 5.0 |
| Very high | 4.8 | 4.5 | 4.3 | 4.3 | 4.3 |
| **Proportion of depressive cases treated in the last year†** | | | | | |
| **Sex** | | | | | |
| Male | 13.3 | 11.9 | 10.7 | 10.6 | 11.1 |
| Female | 15.1 | 14.7 | 16.9 | 16.6 | 15.9 |

Continued

**Table 2** Continued

| | 2014 (%) | 2015 (%) | 2016 (%) | 2017 (%) | 2018 (%) |
|---|---|---|---|---|---|
| **Age** | | | | | |
| 15–34 | 15.7 | 14.8 | 19.7 | 19.2 | 14.7 |
| 35–54 | 17.3 | 16.1 | 18.0 | 15.9 | 13.6 |
| 55–74 | 12.1 | 12.2 | 10.5 | 10.0 | 16.0 |
| 75+ | 7.4 | 8.5 | 6.2 | 10.0 | 12.5 |
| **Area** | | | | | |
| Rural | 7.9 | 7.2 | 7.1 | 5.6 | 7.8 |
| Urban | 17.5 | 18.6 | 21.0 | 18.4 | 16.9 |
| **Wealth index** | | | | | |
| Very low | 5.8 | 5.7 | 5.8 | 3.3 | 6.2 |
| Low | 8.8 | 11.6 | 14.1 | 13.2 | 10.1 |
| Middle | 16.3 | 13.8 | 17.1 | 15.3 | 17.4 |
| High | 20.3 | 22.0 | 24.7 | 14.2 | 20.4 |
| Very high | 27.1 | 31.3 | 31.0 | 38.4 | 26.1 |

Two-stage sample design was taken into account for percentage estimations.
*The analysis considered the total of the Peruvian population.
†An analysis is done by subgroups, considering only people who have depressive symptoms.

and 11.1% in men). However, this was not found to be statistically significant (see table 3).

The individuals in the highest wealth quintile (26.1% in 2018) have a much higher proportion of depressive cases treated than those in the lowest wealth quintile (6.2% in 2018). The probability of receiving treatment was more than five times greater in the highest wealth quintile compared with the lowest wealth quintile

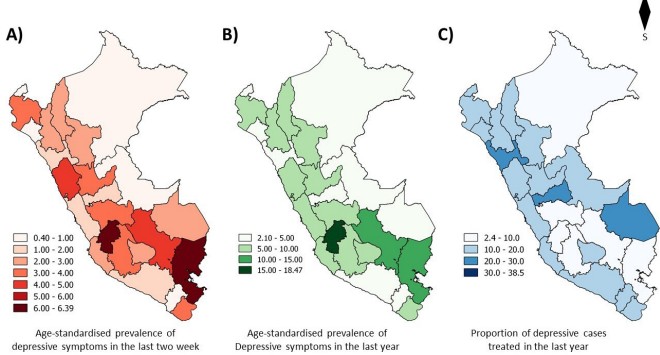

**Figure 2** Age-standardised prevalence of depressive symptoms and proportion of depressive cases treated in Peru by region in 2018. (A) age-standardised prevalence of depressive symptoms in the last 2 weeks from Peru for a region in 2018. (B) Age-standardised prevalence of depressive symptoms in the last year in Peru for a region in 2018. (C) Proportion of depressive cases treated in the last year for the region in 2018. Two-stage sample design was taken into account for percentage estimations. Figure designed by the authors.

**Table 3** Association between receiving treatment and sociodemographic characteristics in people with depressive symptoms in last year, only in 2018

| | Model crude PR (95% CI) | Adjusted model * PR (95% CI) |
|---|---|---|
| **Sex** | | |
| Male | 1 | 1 |
| Female | 1.42 (0.96 to 2.10) | 1.30 (0.88 to 1.92) |
| **Age** | | |
| 15–34 | 1 | 1 |
| 35–54 | 0.92 (0.64 to 1.32) | 0.98 (0.69 to 1.41) |
| 55–74 | 1.09 (0.71 to 1.67) | 1.23 (0.81 to 1.87) |
| 75+ | 0.85 (0.37 to 1.91) | 1.08 (0.49 to 2.39) |
| **Area** | | |
| Rural | 1 | 1 |
| Urban | **2.18 (1.58 to 3.00)\*** | 0.81 (0.48 to 1.37) |
| **Wealth index** | | |
| Very low | 1 | 1 |
| Low | **1.65 (1.05 to 2.60)** | **1.89 (1.06 to 3.38)** |
| Middle | **2.84 (1.78 to 4.50)** | **3.36 (1.73 to 6.53)** |
| High | **3.32 (2.04 to 5.40)** | **3.92 (1.96 to 7.87)** |
| Very high | **4.25 (2.65 to 6.81)** | **5.08 (2.54 to 10.18)** |

Analysis is done by subgroups, considering only people who have depressive symptoms, the complex sampling was used.
Bold values have a significance of p<0.001.
*Adjusted by sex, age, wealth index and area.
PR, prevalence ratio.

(PR=5.82, 95% CI: 2.54 to 10.18, see table 3). No trend of sociodemographic characteristics between 2014 and 2018 was found. At the regional level, the highest proportion of depressive cases treated was on the coast (see figure 2C). The regions with the highest proportion of depressive cases treated were Callao (38.5%) and La Libertad (28.8%). Those with the lowest proportion of depressive cases treated were Puno (4.8%) and Huancavelica (5.5%), which are regions both located in the highlands (online supplementary file 3).

When performing the subanalysis that includes the DSM-5 criteria, the trend results were not different, and the probability of receiving treatment was very similar between each other (14.5% in 2014 to 13.0% in 2018).

## DISCUSSION
### Main results
In Peru, from 2014 to 2018, no changes in prevalence rates (in the 2 weeks prior and over the last year) were found. Similarly, no change in the trends of treatment rates was found. The proportion of people with depressive symptoms receiving treatment was lower in people who live in rural areas and who are of a low level of wealth. Therefore, this situation could be generating a case of inequality in access to treatment in Peru, related to social determinants such as wealth and geographical location.

The results from this study, therefore, provide evidence for a need for an increased commitment and focus on the mental health reforms recently initiated by the Peruvian Ministry of Health. Despite strong evidence of a high prevalence of depression in the country, treatment rates remain low. Along with significant socioeconomic inequalities across the country, it requires an increased allocation of funds and additional resources to prevent and treat depression in the population.

### Comparison with other studies
#### Prevalence of depressive symptomatology
In comparison to other countries in Latin America, Peru has demonstrated a lower (unstandardised) prevalence of depressive symptoms than Brazil (9.7%) while sharing similar levels to those found in Colombia (5.1%).[30 31] These findings have additionally been supported by other studies.[1] Contrastingly, high-income countries such as Australia (9.8%) and the USA (8.4%) have exhibited higher age-standardised prevalences than those found in Peru.[32 33] Other countries have reported that high-income countries, such as the USA and Australia, have a population with a higher proportion of people suffering from non-communicable and chronic diseases. There is evidence that people with chronic and non-communicable diseases have a higher proportion of depressive symptoms,[34] hence the higher prevalence estimate in high-income countries. It should be noted that these studies may be overestimating the proportion of depressive symptoms, as the self-report instruments tend to have higher values than structured diagnostic interviews.[35]

No variation in the prevalence of depressive symptoms was identified in Peru between 2014 and 2018. Our results are consistent with a meta-analysis of 116 epidemiological studies (1990 to 2010)[9] and with other epidemiological studies conducted in Chile (2003 vs 2010) and Germany (1997–1999 vs 2009–2012), where only two measurements were used over a period of several years.[36 37] Contrastingly, three population-based studies found that there has been an increase in depressive symptoms. The first was carried out in the USA and measured prevalence over a period of 11 years (2005 to 2015), the second was conducted for 2 years in Denmark (2000 vs 2006) and the last took place in northeast Germany between 1997–2001 and 2008–2012.[12 13 38] The heterogeneity presented in these results can be explained by the variation in the number of measurements and the amount of time evaluated between the different studies. In order to obtain a stable trend, it is necessary to have consecutive measurements. Although our work only analysed data between 2014 and 2018, it is perhaps the first to report estimates across consecutive years using nationally representative surveys.

It should be noted that the studies, as mentioned earlier, evaluate only depressive symptoms and not major depressive disorders.

## Proportion of depressive cases treated

The treatment gap reported by the Peruvian Ministry of Health (85.9% in 2017) is consistent with our results.[39] However, the treatment gap is higher than those reported in other countries such as the USA (72.4%)[40] or Chile (78.8%).[36] This larger gap could be attributed to the stigma associated with seeking mental healthcare and treatment, preventing patients actually to seek professional attention.[41] Limited resources and a lack of trained health personnel could also play a role.[39 42]

No significant variation was found by year in the proportion of depressive cases treated in Peru. This situation is different from that of USA[43] and the Republic of Georgia,[44] where there has been a significant increase in the proportion of depressive cases treated. The difference could be attributed to the new mental health policies that these countries have implemented and the amount of money they invest in mental healthcare. Peru is undergoing a mental health reform focused on primary care with the implementation of Community Mental Health Centres (CMHCs).[16] The effects at the population level, especially with regards to the proportion of the population with depressive symptoms receiving treatment has not yet been recorded. This could be because it is too early to observe the effects of the CMHCs, as this policy started in 2015 with 22 CMHCs, whereas by the end of 2018, there were 106 CMHCs throughout Peru. Besides, CMHCs are not distributed in a decentralised manner. In the region of Lima, there are only three specialised psychiatric hospitals in Peru and 20% of all CMHCs, and this could explain why the Callao region (with only 147 km² of surface and located very close to Lima) has the highest proportion of depressive cases treated (23.8%). This region, despite its small size, has three CMHCs. Moreover, it is in close proximity to the three psychiatric hospitals in Peru and also other CMHCs. In comparison with Puno, which has the lowest proportion of depressive cases treated (4.8%), which is possibly due to the fact that it only has two CMHCs to supply a territory of 66 997 km². Although largely speculative, these data suggest that the treatment gap could be reduced if CMHCs and other mental health services were widely available in other regions.

## Prevalence and treatment of depressive symptomatology

It could be argued that increasing the proportion of people receiving treatment can help reduce the prevalence of depressive symptoms, but this does not seem to be supported by the evidence. A study conducted in four high-income countries identified that a higher proportion of care is not related to reducing the prevalence of mental health problems.[45] On the other hand, a review suggests that the prevalence of depressive symptoms is reduced if the quality of mental healthcare is improved, waitlists are eradicated, antidepressants are prescribed with greater caution or psychological treatments are opted for, and preventive interventions linked to the community are carried out.[46]

## Relevance in public health

The prevalence of depressive symptoms in Peru was similar to other countries; however, only 14.1% of first-level care centres provide mental health services.[39] This situation represents a public health concern as the lack of access to treatment is considered one of the causes of the treatment gap of mental health experienced by the Peruvian health system.[42] The Peruvian mental health reform seeks, among other objectives, to improve access to mental healthcare. Nonetheless, our results indicate that it is still too early to see the effects of this reform as the treatment gap has not been reduced at the population level in recent years (between 2014 and 2018), at least not in people with depressive symptoms. Despite this, the implementation of CMHCs seems to be a beneficial policy since the health networks that implemented a CMHC increased up to four times the number of mental healthcare consultations, compared with before its implementation.[39] Based on this, we could expect that the implementation of new CMHCs might increase the number of those suffering from mental health problems being attended to, and thereby reduce the treatment gap.

There is evidence that people with a clinical diagnosis of depression are more likely to receive treatment,[47] and the implementation of screening for mental health problems in primary care, this has proven to be very useful in other countries to improve the diagnosis and follow-up.[48 49] However, population-based screening for depressive symptoms—or other mental health problems—has not been included in any Peruvian clinical guidelines.[50 51] Even though CMHCs aim to screen for mental health problems, their capacity is still limited and therefore it is not feasible to cover large populations. Even if other primary care facilities conducted this screening initiative, there is a limited number of psychologists, psychiatrists and primary care physicians who can perform these screening tests.[42] It is therefore necessary for the design and implementation of a population-based screening which will require the involvement of lay people (eg, community health workers) to fill the gap in trained health personnel. Finally, in order for a clinical practice guideline to be successfully implemented and accepted by the patients, they should be involved in the development of these guidelines.[52]

## Strengths and limitations

The present study includes the analysis of five measurements from different years of information collected from nationally representative surveys in the Peruvian population . This is the first study assessing the trends of depressive symptoms in Peru. However, this study has limitations. First, the evaluation period may have been too short (between 2014 and 2018). If prevalence or treatment was to increase, it most likely is small but sustained annually. It would therefore have higher number of measurements (10 or 20 years) in order to be able to identify a trend of higher or a decrease of depression symptoms. Second, although the evaluation of depressive symptoms

was performed using a valid tool,[53] this does not replace an evaluation conducted by a psychiatrist; thus, misclassification may be an issue. Third, another element that could generate bias is the cut-off point used to classify people with depressive symptoms (PHQ-9 scores ≥15). However, when the post hoc analysis was performed using the DSM-5 criteria, the main results were identified as the same (no change in prevalence trends and no change in the proportion of depressive cases treated). Fourth, possible loss mechanisms in the missing data were not evaluated. Fifth, the modified version of the PHQ-9 (in the last year) may have introduced measurement bias. Although evidence of reliability and validity was presented in this study by internal structure, further studies are required to obtain other evidence of validity (ie, relationship to other variables, invariance, sensitivity/specificity). In particular, studies on the measurement validity of the modified version of the PHQ-9 are required. Finally, other important variables that would have allowed a better understanding of the results were not included. For example, type of treatment received, duration of treatment and if the participant has previously received a diagnosis of depression.

## CONCLUSIONS

No significant trend was found in the age-standardised prevalence of depressive symptoms (in the last 2 weeks and the last year) or in the proportion of depressive cases treated. The proportion of depressive cases treated was lower in people with depressive symptoms who lived in rural areas and who were of a low level of wealth. It is therefore necessary to implement policies and interventions in order to reduce the treatment gap and the prevalence of depressive symptoms that can be attributed to social inequalities across the country.

**Acknowledgements** DVZ thanks Helena Kesar and Qory Moscoso for their style review in the final version of the English manuscript.

**Contributors** DVZ conceived and designed the analysis, collected the data, contributed data or analysis tools, performed the analysis and wrote the paper. MCA contributed data or analysis tools and wrote the paper. RMCL conceived and designed the analysis, contributed data or analysis tools, performed the analysis, wrote the paper and helped in supervision of the paper. ABO conceived and designed the analysis, contributed data or analysis tools, performed the analysis, wrote the paper and helped in supervision of the paper.

**Funding** This work was supported by the Strategic Award, Wellcome Trust-Imperial College Centre for Global Health Research (100693/Z/12/Z), Imperial College London Wellcome Trust Institutional Strategic Support Fund (Global Health Clinical Research Training Fellowship) (294834/Z/16/Z ISSF ICL). RMCL was supported by a Wellcome Trust International Training Fellowship (214185/Z/18/Z). The funders had no role in the study design, data collection, analysis, decision to publish or preparation of the manuscript.The authors disclosed the following grant information—Strategic Award, Wellcome Trust-Imperial College Centre for Global Health Research: 100693/Z/12/Z. Imperial College London Wellcome Trust Institutional Strategic Support Fund: 294834/Z/16/Z ISSF ICL. Wellcome Trust International Training Fellowship: 214185/Z/18/Z.

**Map disclaimer** The depiction of boundaries on the map(s) in this article do not imply the expression of any opinion whatsoever on the part of BMJ (or any member of its group) concerning the legal status of any country, territory, jurisdiction or area or of its authorities. The map(s) are provided without any warranty of any kind, either express or implied.

**Competing interests** None declared.

**Patient consent for publication** Not required.

**Ethics approval** The data used in our study are openly accessible to the general public. They do not use any personal identifiers (anonymous) and consequently do not represent an ethical risk for participants. The National Institute of Statistics and Informatics, a Peruvian government organisation, was responsible for the collection of ENDES data. This institution requested the consent of participants to obtain the information required in the survey. The informed consent for the collection of information was taken from each person of legal age (18 years old and above). In the case of minors (17 years old and younger), the request for consent was read to one of their parents or legal guardians to allow the evaluation of the minor.

**Provenance and peer review** Not commissioned; externally peer reviewed.

**Data availability statement** Data are available in a public, open access repository. The database is freely accessible from the website of the National Institute of Statistics of Peru, URL:http://iinei.inei.gob.pe/microdatos/. The information can be obtained by entering the survey query tab and selecting the ENDES data using the health module data. Only cross-sectional information from 2014 to 2018 for the ENDES Health Questionnaire was used.

**ORCID iDs**
David Villarreal-Zegarra http://orcid.org/0000-0002-2222-4764
Rodrigo M Carrillo-Larco http://orcid.org/0000-0002-2090-1856
Antonio Bernabe-Ortiz http://orcid.org/0000-0002-6834-1376

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
