## [Reviewer comments · BMJ Open]

ARTICLE DETAILS

TITLE (PROVISIONAL)	Trends in the prevalence and treatment of depressive symptoms in Peru: A population-based study
AUTHORS	Villarreal-Zegarra, David; Cabrera-Alva, Milagros; Carrillo-Larco, Rodrigo; Bernabe-Ortiz, Antonio

VERSION 1 – REVIEW

REVIEWER	Niina Markkula Helsinki University Central Hospital, Department of Psychiatry, Finland
REVIEW RETURNED	26-Jan-2020

GENERAL COMMENTS	This is a well-written paper and an important topic of study. I found the results interesting and of public health interest. I have some concerns and suggestions about the methodology and some suggestions how to improve the manuscript. Please find them attached. Abstract Design “using secondary data (five-years)” this part is unclear. What do you mean by secondary data? And with five years? Could you add the number of surveys analysed in the objectives, “using xxx nationally representative surveys”? Main outcome measures: PHQ-9 is Patient health questionnaire (not Patients’s) Please mention cut-off of what was considered clinically significant depressive symptoms. Please mention also treatment rate (perhaps self-reported?) as an outcome measure. Results: No evidence of a trend in what? Prevalence? Please specify. “percentage of treatment” could you think of another way to express this? Did you analyse trends in treatment access? Conclusions: this is a matter of taste, but I would mention the lack of trend later, as the second phrase. Strengths and limitations -this is unclear to me, could you clarify? “There is evidence suggesting the use of the cut-off point of 15 in PHQ-9 since the severity of depressive symptoms is consistent with a greater depression that normally requires medication.” What kind of misclassification could the cut-off produce? I think the main limitation is that you did not measure depressive episodes (diagnosis of MDD or MDE) but depressive symptoms, which you acknowledge. In population surveys evaluation by
---

	psychiatrist is not possible but you could use structured interviews that produce diagnostic evaluations such as CIDI or MINI. Introduction  -the topic of treatment is not very well linked with the previous phrase (In addition to this, treatment of depressive symptoms...) -the earlier GBD did not find an increase in the prevalence of depression, but the burden of disease due to depression increased due to changes in population structure (e.g. Ferrari 2013) Ferrari, A. J., Charlson, F. J., Norman, R. E., Flaxman, A. D., Patten, S. B., Vos, T., & Whiteford, H. A. (2013). The epidemiological modelling of major depressive disorder: application for the Global Burden of Disease Study 2010. PLoS One, 8(7). -the GBD 2016 actually found a decrease of -3.6% in the age-standardised rate of depressive disorders from 2006 to 2016. Please change the introduction to reflect this fact, the data has now been cited incorrectly. Methods  -what was the sample size each year? -how was the data collected? mail questionnaire, face to face? -what was the sampling frame? -if the secondary sampling unit was a household, how was the participant within the household selected? did one household member answer for all other members, or all were interviewed individually? -what was the participation rate? this is important in depression studies  -the modified PHQ-9 is interesting. Is there any publication to validate this methodology of anchoring the depressive episode to a troublesome event? -was the ENDES representative at the regional level? -what was the linear model used for? It is unclear from the description. -what statistical method was used to analyse trends? -regarding the subgroup analysis of those filling DSM-5 depression criteria, I would find that you need a score of 2 at least for the core symptoms of depression (meaning sadness or anhedonia more than half of the time), since the diagnostic criteria require these symptoms to be present “most of the day, nearly every day” -DSM-5 not DSM-V -was ethical permit sought and given for this study? Results  -I would find it interesting to present the prevalence rates also for the DSM-5 criteria cases, and compare to the simpler definition -I would present the prevalence rates in more detail, before going into the lack of trends. Discussion  -I would emphasise more the finding of low access to treatment and particularly among socioeconomically disadvantaged groups. Again, lack of trend is not an interesting or surprising finding, in my opinion. Without stressing this finding, it is difficult to understand how this study relates to the “need of an increased commitment and focus on the mental health reforms” (which I agree with). -“changes in trends” perhaps replace with “changes in prevalence rates”
--	--

	-if you need to shorten from somewhere, I think the chapter on gender is not very necessary, the finding of gender difference is not specific to Peru -when comparing to prevalence in other countries you need to take into account that for 12-month prevalence an unvalidated methodology was used in this study (the modified PHQ-9), and in general symptom questionnaires tend to give higher prevalence rates than structured interviews -“No significant variation was found” please add “by year” or similar to clarify -the discussion of the CMHCs and their location is interesting -I disagree with “prevalence of depressive symptoms is very high” - in global comparison it is rather average -could you think of another way to say “percentage of treatment” Tables and figures -the figures are beautiful and nicely illustrate the findings -I would include also some of the tables in the main publication, not as supplements, perhaps table 2?
--	--

REVIEWER	Tao Chen Liverpool School of Tropical Medicine
REVIEW RETURNED	20-Feb-2020

GENERAL COMMENTS	David Villarreal-Zegarra et al performed an analysis to estimate the trends of depression prevalence and treatment on the basis of an open, nationally representative surveys data in Peru. This research was designed and analysed in a proper way. I had some minor comments below. 1, Some analyses, such as changing the cutoff point of PHQ-9, subanalysis that includes the DSM-V criteria, were mentioned in the main text but were not included in the supplement for further assessment. 2, Poisson model was used but did not specify for which outcome. does it mean for all variables comparison or only for prevalence? 3, “For the comparisons between groups, the Pearson χ^2 test was used to compare those receiving treatment and the sociodemographic characteristics in people with depressive symptoms.” I could not find the mentioned comparisons. 4, the standardised prevalence is very important for inter-countries comparison however, unstandardised prevalence is also vital, especially to the policymaker who use these data to estimate the disease burden in Peru.
---

REVIEWER	Darío Moreno-Agostino King's College London, United Kingdom
REVIEW RETURNED	20-Feb-2020

GENERAL COMMENTS	This paper explores the potential change in the prevalence of depression and its treatment in a nationally representative Peruvian sample. The work is interesting, well described, and relevant. It uses information from different time points with similar implementation methods which facilitates the comparability across the cross-sections. Additionally, is conducted on a low- and middle-income country, which may help reducing the evidence gap in mental health regarding these countries.
---

Nevertheless, there are several points that, if addressed, could substantially improve the strength and clarity of the manuscript:

General comments:

- Since the manuscript is focused on the trends, it would be nice to include some information on some of the other time points considered, and not only on 2018. Including at least the estimates for 2014 could clarify this. This should be done in the abstract and in the results section.
- Throughout the text, the authors mention that they are covering a period of 5 years, but this is not completely correct: they are covering a period of 4 years with 5 time points.
- The manuscript could benefit from a thorough revision of the English. This is especially true in the Discussion section, that also feels less carefully written.

Methods:

- The first paragraph of the “Variables” subsection seems redundant to the following two paragraphs. In this first paragraph, I understand the authors try to say that, in addition to the original PHQ-9 (which covers the last two weeks), a modified version was also administered to assess the experience of depressive symptoms in the last 12 months. This is not clear in the text; rather, the reader has to figure this out.
- The bit on the “age-standardized prevalence” (lines 44-47 of page 7) should go to Statistical analysis.
- Regarding the 12-month modification of the PHQ-9, participants are asked about “an event in the last 12 months in which they had discomfort or problems”. Although I am aware that this was not under the control of the authors, I am wondering whether it should have been more adequate to ask for the event in which they felt the most discomfort, in order to better capture the worse moment mood-wise. I am not certain that this measure is valid to assess the presence of depressive symptomatology in the last 12 months, in the sense that it could leave out people with worse episodes due to the wording of the question.
- The authors mention they considered socioeconomic status (SES) by means of measures of wealth. I suggest referring to this as wealth, since SES cannot be captured only by wealth (please, check Braveman et al 2005: <https://jamanetwork.com/journals/jama/article-abstract/202015>).
- In the same line, it would be wonderful to consider in the models the educational attainment, if possible.
- I assume the quintiles of this wealth index are performed within each cross-section. I suggest specifying this in the text.
- It would be nice to know how the “analyses performed took into consideration the complex sampling design of the study”.
- Also, it could be worth clarifying the aim of the “post hoc analysis”.

Results:

- Mann-Kendall Tau test may not be sensitive to detect change with these few observations (although the graphic inspection of the results also suggests that there seem not to be any monotonic trend). Nevertheless, I suggest the authors to report the statistic (tau) and not only its significance, so the reader can know what those significances mean.
- The same applies to what I think is the significance of the chi-square test (line 43-44, page 9). The chi-square and degrees of freedom should be specified here, along with the significance.

	 - The last line of the results is repeated in the text. - Table 1: why are there such differences in the percentage of people at the highest and lowest wealth quintiles in 2015 and 2016? - Table 1: the note is confusing. Can it be clarified? - Supplement 1: in my experience, it is very peculiar that the excluded cases due to missing are due to missing data on sex and age, and on the contrary there are no missing data on wealth (or SES, as the authors include this index). Can the authors confirm this is the case? Discussion:  - The authors include Brazil as a high-income country, when it is categorised as a low- and middle-income country. - I find the following sentence a bit confusing: "Sex is a factor that could be influencing the difference between groups because women tend to have an increased risk of having depressive symptoms". By the position in the text, it is supposed to be somehow an explanation, but it is rather a circular description of the very results found. I suggest rewording it. - Line 45, page 10 is repeated. - The authors mention that there are two papers that show an increase in depressive symptoms. If the authors are limiting this statement to studies focused on symptomatology rather than MDE/MDD (there are many studies with this operationalisation that find increasing trends), they should specify that here. Nevertheless, there are other studies (e.g. https://www.ncbi.nlm.nih.gov/pubmed/24462337, https://www.ncbi.nlm.nih.gov/pubmed/25500698) that also find an increasing trend considering symptomatology. - Line 14, page 24: do the authors refer to "depressive symptoms over time"? - Lines 36-37, page 24: this statement seems too strong for the relatively little support that it receives, and the following counterargument that is included. I suggest including more evidence in support or tempering it. - Line 33, page 23: the authors say "any misclassification present in the study may be non-differential". Are they trying to say it may be constant across time points? This is also included in the strengths and limitations of the study.
--	---

VERSION 1 – AUTHOR RESPONSE

Reviewer(s)' Comments to Author:

Reviewer: 1

This is a well-written paper and an important topic of study. I found the results interesting and of public health interest. I have some concerns and suggestions about the methodology and some suggestions on how to improve the manuscript. Please find them attached.

Abstract

Design "using secondary data (five-years)" this part is unclear. What do you mean by secondary data? And with five years? Could you add the number of surveys analyzed in the objectives, "using xxx nationally representative surveys"?

Reply: The design section was changed to: " Design: A secondary analysis was conducted using five consecutive nationally representative surveys conducted from 2014-2018, including a total of 166,290 individuals."

Main outcome measures: PHQ-9 is Patient health questionnaire (not Patients)

Please mention the cut-off of what was considered clinically significant depressive symptoms. Please also refer to the treatment rate (perhaps self-reported?) as an outcome measure.

Reply: We have added the reviewer's suggestions as follows:

"Two versions of the Patient Health Questionnaire (PHQ-9) that focused on the presence of depressive symptoms in the last two weeks and the last year were used. Scores of ≥ 15 were used as the cut-off point in both versions of the PHQ-9 to define the presence of depressive symptoms. In addition, the treatment rate was based on the proportion of individuals with depressive symptoms in the last year who self-reported having received specific treatment for these symptoms. The age-standardized prevalence was estimated using the World Health Organization population as the reference population."

Results: No evidence of a trend in what? Prevalence? Please specify. "percentage of treatment" could you think of another way to express this? Did you analyze trends in treatment access?

Reply: It was specified in the abstract in the results section:

"A total of 161,061 participants were included. There was no evidence of a change in age-standardized prevalence rates of depressive symptoms at two weeks (2.6% in 2014 to 2.3% in 2018), or in the last year (6.3% in 2014 to 6.2% in 2018). No change was found either in the proportion of depressive cases treated in the last year (14.6% in 2014 to 14.4% in 2018). This latter proportion was lower in people with a low level of wealth."

Conclusions: This is a matter of taste, but I would mention the lack of trend later, as the second phrase.

Reply: We have added the reviewer's suggestions as follows:

"One in fifty people exhibited depressive symptoms during the last two weeks and three in fifty in the last year. Also, one in seven individuals with depressive symptoms reported receiving treatment. However, there was no evidence of a decreasing or increasing trend."

Strengths and limitations

-this is unclear to me, could you clarify? "There is evidence suggesting the use of the cut-off point of 15 in PHQ-9 since the severity of depressive symptoms is consistent with a greater depression that normally requires medication." What kind of misclassification could the cut-off produce?

I think the main limitation is that you did not measure depressive episodes (diagnosis of MDD or MDE) but depressive symptoms, which you acknowledge. In population surveys, evaluation by a psychiatrist is not possible, but you could use structured interviews that produce diagnostic assessments such as CIDI or MINI.

Reply: The strengths and limitations section were modified:

- This analysis included information from five nationally representative surveys of the Peruvian population.
- Our findings were limited as depression assessment was not used psychiatric evaluations or structured interviews; so, only depressive symptoms could be identified.
- Only five years were evaluated and perhaps more time may be required to identify a significant trend.

Introduction

-the topic of treatment is not very well linked with the previous phrase (In addition to this, treatment of depressive symptoms...)

Reply: The wording of that paragraph was reviewed and modified:

"On the other hand, the fact that a person with depressive symptoms can access treatment also depends on accessibility to health systems. A systematic review identified that lack of human resources, centralization of the health system, and integration in primary care, are barriers to receiving appropriate treatment of depressive symptoms (8)."

-the earlier GBD did not find an increase in the prevalence of depression, but the burden of disease due to depression increased due to changes in population structure (e.g., Ferrari 2013)

Ferrari, A. J., Charlson, F. J., Norman, R. E., Flaxman, A. D., Patten, S. B., Vos, T., & Whiteford, H. A. (2013). The epidemiological modeling of major depressive disorder: application for the Global Burden of Disease Study 2010. *PLoS One*, 8(7).

Reply: This section was modified:

"Nevertheless, the Global Burden of Disease Studies assessing trends in previous years (10, 11), found a decrease of 4.9% in the age-standardized rate of depressive disorders from 2006 to 2016. On the contrary, other longitudinal studies showed an increase in the prevalence of major depressive disorder (12, 13)."

-the GBD 2016 found a decrease of -3.6% in the age-standardized rate of depressive disorders from 2006 to 2016. Please change the introduction to reflect this fact; the data has now been cited incorrectly.

Reply: This section was modified. The difference identified in Figure 1 of the paper of GBD 2016 was -4.9%, so this figure was placed.

"Currently, there is no consensus on the trends of the prevalence of depressive symptoms. A meta-analysis of 116 epidemiological investigations during the years 1990 and 2010 did not reveal changes in the prevalence of major depressive disorder (9). Nevertheless, the Global Burden of Disease Studies assessing trends in previous years (10, 11), found a decrease of 4.9% in the age-standardized rate of depressive disorders from 2006 to 2016. On the contrary, other longitudinal studies showed an increase in the prevalence of major depressive disorder (12, 13). These mixed results can be attributed to inter-country variations of several demographic and socioeconomic factors, as well as the different criteria used to define depression in research studies. Due to the lack of global consensus, country-specific estimates and trends are significant and can inform local policies and guidelines."

Methods

-what was the sample size each year?

Reply: The number of participants in each year was included. This was placed in the method section in the subsection on participants:

"The number of participants evaluated by the ENDES in each year is about 30,000 (see Table 1)."

-how was the data collected? Mail questionnaire, face to face?

Reply: It was noted that the data collection was face-to-face and that each participant was interviewed for the data. This was placed in the method section in the study design subsection:

"The data collection, in each of the years, used a face-to-face approach and started in February or March, while the completion ended in December (17, 18)."

-what was the sampling frame?

Reply: The information was added:

"The sampling used was probabilistic in two stages and representative at the national and regional levels. The sampling frame in the first stage was the selection of primary sampling units (clusters) based on information from the last census conducted in Peru. In the second stage, the selection of secondary sampling units (households) was carried out based on the information from the cartographic updates and the register of buildings and households made previously (17)."

-if the secondary sampling unit was a household, how was the participant within the household selected? Did one household member answer for all other members, or all were interviewed individually?

Reply: This information was added:

"Information on the dates of birth of all household members and the order in which these data were collected was used for the selection of participants. Only one participant aged 15 years or older was selected from each household. The participant with the closest birthday to the evaluation date was selected. In the event of a tie between two or more participants' birthdays (i.e. same birth day), the participant whose data was collected first was selected (19)."

-what was the participation rate? this is important in depression studies

Reply: The participation rate varied between 95.7% in 2014 to 97.42% in 2018. This was placed in the results section in the participant's subsection:

"The participation rate varied between 95.7% in 2014 and 97.4% in 2018."

-the modified PHQ-9 is interesting. Is there any publication to validate this methodology of anchoring the depressive episode to a troublesome event?

Reply: At this time, there is no single publication that validates this modified version. However, it was considered appropriate for our manuscript to present evidence of the validity and reliability of the revised version of the PHQ-9 for the assessment of depressive symptoms in the last year (2018 only). This was placed in the results section and in supplement 2:

"The measurement properties of the modified version of the PHQ-9 to assess depressive symptoms during the last year were evaluated before the primary analyses. The modified version of the PHQ-9 was identified as having evidence of validity by confirmatory factor analysis (CFI=0.92; RMSEA=0.05) and evidence of reliability by internal consistency ($\omega=0.87$; $\alpha=0.86$). More information on the factorial analysis of the modified version of the PHQ-9 is presented in supplement 2."

-was the ENDES representative at the regional level?

Reply: Yes, the ENDES is representative at the national and regional levels. This was placed in the method section in the study design subsection:

"From 2014 onwards, the data collected on mental health was nationally and regional representative. Our study, therefore, decided to use data collected by ENDES between the years 2014 and 2018."

-what was the linear model used for? It is unclear from the description.

Reply: The linear model was performed to compare sociodemographic characteristics in people receiving treatment for depressive symptoms in the last year:

"A sub-analysis was performed to compare sociodemographic characteristics in people receiving treatment for depressive symptoms in the last year. The Pearson χ^2 test was used to make these comparisons (see Supplement 3). Crude and adjusted models were created using a generalized linear model, assuming a Poisson distribution, link log, and robust variance was used as suggested in the literature (25). Prevalence ratios (PR) and 95% CI were reported."

-what statistical method was used to analyze trends?

Reply: Initially, the Mann-Kendall Tau test had been used. However, due to the third reviewer's commentary, it was considered more appropriate to use the score test for trend (tabodds in Stata) adjusting for age-standardized categories. For both the Mann-Kendall test and the tabodds test, the results obtained were similar since no bias in the odds ratios is identified. This information was modified in the abstract, method, results, and discussion sections.

"The analysis was conducted at both country and regional levels. A 95% confidence interval (95% CI) was calculated for the prevalence at the regional and national levels. The trend over time was age-

standardized and evaluated using the score test for trend; for this, the year 2014 was used as the reference category."

-regarding the subgroup analysis of those filling DSM-5 depression criteria, I would find that you need a score of 2 at least for the core symptoms of depression (meaning sadness or anhedonia more than half of the time) since the diagnostic criteria require these symptoms to be present "most of the day, nearly every day."

Reply: The approach was changed to:

"An additional post hoc analysis was performed, which conditions that participants with depressive symptoms must meet the DSM-5 criteria. This analysis aims to identify whether using the clinical criteria proposed by DSM-5 can alter the results (higher or lower) or be equivalent to our main results, which use a score of ≥ 15 in the PHQ-9. For this sub-analysis participants must have feelings of sadness or anhedonia (have a score of two or more on items 1 and 2 of the PHQ-9, i.e., "more than half the days" and "nearly every day") and at least five of the other seven indicators (have a score of at least one on five of the other seven items of the PHQ-9)."

The analyses were re-run but did not change the trend results.

-DSM-5 not DSM-V

Reply: This change was made throughout the text.

-was ethical permit sought and given for this study?

Reply: The data used in our study are openly accessible to the general public and do not use any personally identifiable information (anonymous), and consequently does not represent an ethical risk for participants. The paragraph in the subsection on ethics was amended:

"The data used in our study are openly accessible to the general public. They do not use any personal identifier (anonymous) and consequently do not represent an ethical risk for participants. The National Institute of Statistics and Informatics, a Peruvian government organization, was responsible for the collection of ENDES data. This institution requested the consent of participants to obtain the information required in the survey. For each person of legal age (18 years old and above), the informed consent for the collection of information was taken. In the case of minors (17 years old and younger), the request for consent was read to one of their parents or legal guardians to allow the evaluation of the minor."

Results

-I would find it interesting to present the prevalence rates also for the DSM-5 criteria cases and compare to the more straightforward definition

Reply: The results were very similar to those already reported. They are added in the results section: "Trend results were not different when DSM-5 criteria were used. Thus, the age-standardized prevalence was very similar for both depressive symptoms in the last two weeks (3.4% in 2014 to 3.3% in 2018) and in the last year (6.8% in 2014 to 6.8% in 2018)."

-I would present the prevalence rates in more detail before going into the lack of trends.

Reply: We consider this a good suggestion. However, the document is close to exceeding 5000 words. So, adding more information could make it very long. So, we have decided not to add more information. However, the information requested by the reviewer is in Table 2 and Figure 1.

Discussion

-I would emphasize more the finding of low access to treatment and particularly among socioeconomically disadvantaged groups. Again, a lack of trend is not an interesting or surprising finding, in my opinion. Without stressing this finding, it is difficult to understand how this study relates to the "need of an increased commitment and focus on the mental health reforms" (which I agree with).

Reply: Aspects related to the low proportion of people receiving treatment were emphasized. This information was noted in the subsection "Main results":

"In Peru, from 2014 to 2018, no changes in prevalence rates (in the two weeks prior and over the last year) were found. Similarly, no change in the trends of treatment rates was found. The proportion of people with depressive symptoms receiving treatment was lower in people who live in rural areas and who are of a low level of wealth. Therefore, this situation could be generating a case of inequality in access to treatment in Peru, related to social determinants such as wealth and geographical location. The results from this study, therefore, provide evidence for a need for an increased commitment and focus on the mental health reforms recently initiated by the Peruvian Ministry of Health. Despite strong evidence of a high prevalence of depression in the country, treatment rates remain low. Along with significant socioeconomic inequalities across the country, this requires an increased allocation of funds and additional resources to prevent and treat depression in the population."

-"changes in trends" perhaps replace with "changes in prevalence rates."

Reply: It was modified:

"In Peru, from 2014 to 2018, no changes in prevalence rates (in the two weeks prior and over the last year) were found. Similarly, no change in the trends of treatment rates was found."

-if you need to shorten from somewhere, I think the chapter on gender is not very necessary, the finding of gender difference is not specific to Peru

Reply: We agree and have deleted the paragraph accordingly.

-when comparing to prevalence in other countries you need to take into account that for 12-month prevalence an unvalidated methodology was used in this study (the modified PHQ-9), and in general symptom, questionnaires tend to give higher prevalence rates than structured interviews

Reply: Within the manuscript it was added that questionnaires tend to give higher prevalence rates than structured interviews:

"It should be noted that these studies may be overestimating the proportion of depressive symptoms. As the self-report instruments tend to have higher values than structured diagnostic interviews (34)."

Despite of this, we used a modified version of the PHQ-9 to assess the presence of depressive symptoms in the last year, and this may bias our findings. Therefore, a sub-analysis was performed to validate the instrument by means of confirmatory factor analysis (evidence of validity by internal structure) and with internal consistency coefficients (reliability). This new information can be seen in supplement 2 and the result:

"The measurement properties of the modified version of the PHQ-9 to assess depressive symptoms during the last year were evaluated before the primary analyses. The modified version of the PHQ-9 was identified as having evidence of validity by confirmatory factor analysis (CFI=0.92; RMSEA=0.05) and evidence of reliability by internal consistency ($\omega=0.87$; $\alpha=0.86$). More information on the factorial analysis of the modified version of the PHQ-9 is presented in supplement 2."

-"No significant variation was found" please add "by year" or similar to clarify

Reply: This paragraph was modified:

"No significant variation was found by year in the proportion of depressive cases treated in Peru."

-the discussion of the CMHCs and their location is interesting

Reply: thanks

-I disagree with "prevalence of depressive symptoms is very high" - in global comparison it is rather average

Reply: This modification was made in the subsection "Relevance in public health":

"The prevalence of depressive symptoms in the last year in Peru was similar to other countries; however, only 14.1% of first-level care centers provide mental health services (38). This situation represents a public health concern as the lack of access to treatment is considered one of the causes of the treatment gap of mental health experienced by the Peruvian health system (41)."

-could you think of another way to say, "percentage of treatment."

Reply: Changed to "Proportion of depressive cases treated" in all text.

Tables and figures

-the figures are beautiful and nicely illustrate the findings

Reply: Thanks

-I would also include some of the tables in the primary publication, not as supplements, perhaps table 2?

Reply: The table in supplement 2 was added to the main manuscript. The new table is named table 3.

Reviewer: 2

Please leave your comments for the authors below

David Villarreal-Zegarra et al performed an analysis to estimate the trends of depression prevalence and treatment based on an open, nationally representative survey data in Peru. This research was designed and analyzed correctly. I had some minor comments below.

1. Some analyses, such as changing the cut-off point of PHQ-9, subanalysis that includes the DSM-V criteria, were mentioned in the main text but were not included in the supplement for further assessment.

Reply: This was added within the main text in the results section:

"Trend results were not different when DSM-5 criteria were used. Thus, the age-standardized prevalence was very similar for both depressive symptoms in the last two weeks (3.4% in 2014 to 3.3% in 2018) and in the last year (6.8% in 2014 to 6.8% in 2018)."

2. The Poisson model was used but did not specify for which outcome. Does it mean for all variable's comparison or only for prevalence?

Reply: It has been modified in the subsection on statistical analysis in the methods section. It was pointed out that this subanalysis was only done in people who had been treated for depressive symptoms in the last year:

"A sub-analysis was performed to compare sociodemographic characteristics in people receiving treatment for depressive symptoms in the last year. The Pearson χ^2 test was used to make these comparisons (see Supplement 3). Crude and adjusted models were created using a generalized linear model, assuming a Poisson distribution, link log, and robust variance was used as suggested in the literature (25). Prevalence ratios (PR) and 95% CI were reported."

3. "For the comparisons between groups, the Pearson χ^2 test was used to compare those receiving treatment and the sociodemographic characteristics in people with depressive symptoms." I could not find the mentioned comparisons.

Reply: The comparisons mentioned are in supplement 3. This information is noted within the main text:

"A higher proportion of women self-reporting for treatment was found when compared to men (only 2018, 15.9% in women and 11.1% in men). However, it was not found to be statistically significant (see Table 3)."

4. The standardized prevalence is significant for inter-countries comparison; however, unstandardized prevalence is also vital, especially to the policymaker who uses these data to estimate the disease burden in Peru.

Reply: The un-standardized prevalence information had already been noted in the text in the results section:

"On the other hand, in 2018, the un-standardized prevalence of depressive symptoms in the last two weeks was 2.7% (95% CI: 2.4%-2.9%) and in the last year was 6.2% (95% CI: 5.8%–6.6%) and therefore showed little to no difference from the age-standardized prevalence results."

Reviewer: 3

This paper explores the potential change in the prevalence of depression and its treatment in a nationally representative Peruvian sample. The work is interesting, well described, and relevant. It uses information from different time points with similar implementation methods, which facilitates the comparability across the cross-sections. Additionally, it is conducted in a low- and middle-income country, which may help to reduce the evidence gap in mental health regarding these countries.

Nevertheless, there are several points that, if addressed, could substantially improve the strength and clarity of the manuscript:

General comments:

- Since the manuscript is focused on the trends, it would be nice to include some information on some of the other time points considered and not only in 2018. Including at least the estimates for 2014 could clarify this. This should be done in the abstract and the results section.

Reply: The abstract was modified:

"A total of 161,061 participants were included. There was no evidence of a change in age-standardized prevalence rates of depressive symptoms at two weeks (2.6% in 2014 to 2.3% in 2018), or in the last year (6.3% in 2014 to 6.2% in 2018). No change was found either in the proportion of depressive cases treated in the last year (14.6% in 2014 to 14.4% in 2018). This latter proportion was lower in people with a low level of wealth."

- Throughout the text, the authors mention that they are covering 5 years, but this is not entirely correct: they are covering 4 years with 5-time points.

Reply: It was modified throughout the text, indicating that the results correspond to 2014-2018.

To better explain our results, information was added about the months in which the ENDES information was collected (method section, design topic):

"The data collection, in each of the years, used a face-to-face approach and started in February or March, while the completion ended in December (17, 18). Data collection was face-to-face, with each participant evaluated being interviewed."

- The manuscript could benefit from a thorough revision of the English. This is especially true in the Discussion section, which also feels less carefully written.

Reply: A complete revision of the English in the manuscript was carried out.

Methods:

- The first paragraph of the "Variables" subsection seems redundant to the following two paragraphs. In this first paragraph, I understand the authors try to say that, in addition to the original PHQ-9 (which covers the last two weeks), a modified version was also administered to assess the experience of depressive symptoms during the previous 12 months. This is not clear in the text; instead, the reader has to figure this out.

Reply: The first paragraph was deleted. The differentiation between the original version of the PHQ-9 (lasts two weeks) and the version modified to assess depressive symptoms in the last year was better explained:

"A modified version of the PHQ-9 was used to assess depressive symptoms over the last year to help identify the occurrence of depressive symptoms over a more extended time. This version of the PHQ-9 was used in ENDES 2014-2018 to evaluate depressive symptoms experienced at some point in the past 12 months."

- The bit on the "age-standardized prevalence" (lines 44-47 of page 7) should go to Statistical analysis.

Reply: The information was moved to the Statistical Analysis subsection in the methods section.

- Regarding the 12-month modification of the PHQ-9, participants are asked about "an event in the last 12 months in which they had discomfort or problems". Although I am aware that this was not under the control of the authors, I am wondering whether it should have been adequate to ask for the event in which they felt the most discomfort in order to capture the worse moment mood-wise better. I am not sure that this measure is valid to assess the presence of depressive symptomatology in the last 12 months, in the sense that it could leave out people with worse episodes due to the wording of the question.

Reply: A confirmatory factor analysis was performed on the modified version of PHQ-9 (in the last year). The revised version of the PHQ-9 for the assessment of depressive symptoms in the previous year is considered to present evidence of validity and reliability:

"The measurement properties of the modified version of the PHQ-9 to assess depressive symptoms during the last year were evaluated before the primary analyses. The modified version of the PHQ-9 was identified as having evidence of validity by confirmatory factor analysis (CFI=0.92; RMSEA=0.05) and evidence of reliability by internal consistency ($\omega=0.87$; $\alpha=0.86$). More information on the factorial analysis of the modified version of the PHQ-9 is presented in supplement 2."

The steps that were performed for the analysis are added in the method section:

"To evaluate the measurement properties of the modified version of the PHQ-9 that collects information on depressive symptoms in the last year, a subanalysis was conducted (see Supplement 2). A confirmatory factor analysis was performed to evaluate the validity of the modified version of the PHQ-9, considering the ordinal nature of the items and using the estimate of weighted least squares means and variance adjusted (26). It was considered that the instrument presents sufficient evidence of validity if it reaches optimum values in the different goodness-of-fit indices. The Comparative Fit Index (CFI) and the Tucker-Lewis Index (TLI) were used; these indices must be greater than 0.90 to be considered adequate levels (27). The Root Mean Square Error of Approximation (RMSEA) with a confidence interval of 90% and the Standardized Root Mean Square Residual (SRMR) were also used, both indices considering fair values lower than 0.08 (27). On the other hand, reliability was evaluated by the internal consistency coefficient of alpha and omega. Both coefficients consider that adequate levels of reliability are reached if they score higher than 0.80 (28)."

- The authors mention they considered socioeconomic status (SES) utilizing measures of wealth. I suggest referring to this as wealth since SES cannot be captured only by wealth (please, check Braveman et al 2005: <https://jamanetwork.com/journals/jama/article-abstract/202015>).

Reply: It is considered relevant to change socioeconomic level by level of wealth. These changes were made throughout the manuscript.

- In the same line, it would be pleasant to consider in the models the educational attainment, if possible.

Reply: Unfortunately, we did not include information on the educational attainment, since it was not possible to integrate it when putting together the different databases of the different years of evaluation.

- I assume the quintiles of this wealth index are performed within each cross-section. I suggest specifying this in the text.

Reply: This information was specificity in the topic of "Other variables" in the subsection of variables in the section of method:

"The wealth level of participants was defined in quintiles (very low, low, middle, high, and very high) based on a wealth index available in the ENDES (23). This index is calculated by using the availability of goods and services, the housing characteristics that the participants report having (17), and the quintiles of wealth. The index is performed within each cross-section (each year). The calculation of this index can be found in Rutstein and Johnson (23)."

- It would be nice to know how the "analyses performed took into consideration the complex sampling design of the study."

Reply: The information in the subsection on statistical analysis was modified:

"All the analyses performed considered the design by a complex sampling of the ENDES, and the analyses performed were adjusted based on the weight factor provided by each ENDES assessment year. Adjustments were made with the STATA command "svy" for all analyses, except for the factor analysis (sub-analysis) where the "lavaan" and "lavaan.survey" package was used in R."

- Also, it could be worth clarifying the aim of the "post hoc analysis."

Reply: The aim of the post hoc analysis is specified in the subsection on statistical analysis:

"An additional post hoc analysis was performed, which conditions that participants with depressive symptoms must meet the DSM-5 criteria. This analysis aims to identify whether using the clinical criteria proposed by DSM-5 can alter the results (higher or lower) or be equivalent to our main results, which use a score of ≥ 15 in the PHQ-9. For this sub-analysis participants must have feelings of sadness or anhedonia (have a score of two or more on items 1 and 2 of the PHQ-9, i.e., "more than half the days" and "nearly every day") and at least five of the other seven indicators (have a score of at least one on five of the other seven items of the PHQ-9)."

Results:

- Mann-Kendall Tau test may not be sensitive to detect change with these few observations (although the visual inspection of the results also suggests that there seem not to be any monotonic trend). Nevertheless, I recommend the authors report the statistic (tau) and not only its significance, so the reader can know what those significances mean.

Reply: The analysis was modified. The score test for trend (tabodds in Stata) adjusting for age-standardized categories is preferred because it considers the sample size:

"The trend over time was age-standardized and evaluated using the score test for trend; for this, the year 2014 was used as the reference category."

However, the results were not changed:

"In Peru, from 2014 to 2018, no changes in prevalence rates (in the two weeks prior and over the last year) were found. Similarly, no change in the trends of treatment rates was found."

- The same applies to what I think is the significance of the chi-square test (line 43-44, page 9). The chi-square and degrees of freedom should be specified here, along with the significance.

Reply: It was modified, the p-value was removed. It was placed in the text that is not significant, and supplement 3 was pointed out where you can see the PR value (to which we originally wanted to refer):

"A higher proportion of women self-reporting for treatment was found when compared to men (only 2018, 15.9% in women and 11.1% in men). However, it was not found to be statistically significant (see Table 3)."

- The last line of the results is repeated in the text.

Reply: It was modified:

"When performing the subanalysis that includes the DSM-5 criteria, the trend results are not modified, and the probability of receiving treatment was very similar (14.5% in 2014 to 13.0% in 2018)."

- Table 1: why are there such differences in the percentage of people at the highest and lowest wealth quintiles in 2015 and 2016?

Reply: We have looked for a reason for the possible variation in the wealth index in 2015 and 2016. However, we have not identified a study that can explain it. We have re-run the analyses and get the same results, so we consider it a variation of the ENDES data itself.

- Table 1: the note is confusing. Can it be clarified?

Reply: The note in Table 1 was clarified:

"Note: Two-stage sample design was taken into account for percentage estimations."

- Supplement 1: in my experience, it is very peculiar that the excluded cases due to missing are due to missing data on sex and age, and on the contrary, there are no missing data on wealth (or SES, as the authors include this index). Can the authors confirm this is the case?

Reply: We confirm this is the case. We also found it interesting, but there were no missing observations in wealth.

The ENDES collects information on different variables (i.e. index wealth, fertility, health, household). The first questionnaire that is applied is the household questionnaire in which the members of the household are identified, the wealth index, and whether the household has persons eligible for the different questionnaires of the ENDES. Once this questionnaire is completed, the health questionnaire is applied to participants. This is why all participants have the data on the wealth index and there is no missing information.

Discussion:

- The authors include Brazil as a high-income country when it is categorized as a low- and middle-income country.

Reply: Brazil was removed from line 6 of the subsection of "Comparison with other studies" since Brazil is not considered a high-income country.

- I find the following sentence a bit confusing: "Sex is a factor that could be influencing the difference between groups because women tend to have an increased risk of having depressive symptoms." By the position in the text, it is supposed to be somehow an explanation, but it is instead a circular description of the very results found. I suggest rewording it.

Reply: This paragraph was deleted at the suggestion of a reviewer.

- Line 45, page 10, is repeated.

Reply: Line 45 on page 10, located within the sub-section "Prevalence of depressive symptomatology," was deleted because it is repeated.

- The authors mention that two papers show an increase in depressive symptoms. If the authors are limiting this statement to studies focused on symptomatology rather than MDE/MDD (there are many studies with this operationalization that find increasing trends), they should specify that here. Nevertheless, there are other studies (e.g., <https://www.ncbi.nlm.nih.gov/pubmed/25500698>) that also see a rising trend considering symptomatology.

Reply: It was specified that it only focuses on depressive symptoms:

"It should be noted that the studies, as mentioned earlier, evaluate only depressive symptoms and not major depressive disorders."

- Line 14, page 24: do the authors refer to "depressive symptoms over time"?

Reply: We couldn't find the sentence the reviewer is referring to.

- Lines 36-37, page 24: this statement seems too strong for the relatively little support that it receives, and the following counterargument that is included. I suggest including more evidence in favor of tempering it.

Reply: This paragraph was modified in the subsection on strengths and limitations (in discussion):

"Third, another element that could generate bias is the cut-off point used to classify people with depressive symptoms (PHQ-9 scores ≥ 15). However, when the post hoc analysis was performed using the DSM-5 criteria, the main results were identified as the same (no change in prevalence trends and no proportion of treatment)."

- Line 33, page 23: the authors say, "any misclassification present in the study may be non-differential." Are they trying to say it may be constant across time points? This is also included in the strengths and limitations of the study.

Reply: This paragraph was modified in the subsection on strengths and limitations (in discussion):

"Second, although the evaluation of depressive symptoms was performed using a valid tool (52), this does not replace an evaluation conducted by a psychiatrist; thus, misclassification may be an issue."

VERSION 2 – REVIEW

REVIEWER	Niina Markkula Helsinki University and Helsinki University Hospital
REVIEW RETURNED	17-Apr-2020

GENERAL COMMENTS	The authors have done an excellent job responding to the comments and improved the manuscript greatly. Well done! I have no further comments.
---

REVIEWER	Tao Chen Liverpool School of Tropical Medicine
REVIEW RETURNED	16-Apr-2020

GENERAL COMMENTS	The article has been improved a lot. I have no further comments.
--

REVIEWER	Darío Moreno-Agostino King's College London, United Kingdom
REVIEW RETURNED	21-Apr-2020

GENERAL COMMENTS	The authors have adequately addressed most of the comments and suggestions, thus enhancing the article's quality. Nevertheless, additional changes should be made to clarify some aspects: GENERAL COMMENTS: - The article could still benefit from an additional revision of the English ABSTRACT:
--

- The "Conclusions" in the abstract merely contain results. There is no interpretation of them and/or their implications.

METHODS:

- Page 34, line 22: the commas should go after the quotation marks

- Page 32, line 31: "and the quintiles of wealth". Is this a typo? Otherwise, what are these "quintiles of wealth" that, according to this paragraph, are used in the index that will be recoded into quintiles later on?

- Statistical methods: in general, this section would benefit from a clearer presentation, starting with the main analyses, clearly outlined, then the sub-analyses, and finally the statistical software used to perform them.

- In the same line, the main analysis should correspond, according to the aims of the study, to the trend analysis. The authors have changed the analytical approach for these main analyses, but the new strategy to analyse the trends is not clear.

- The Supplement 3 is included in the text before the Supplement 2. This should be changed to reflect the order of appearance.

- The authors state that "It was considered that the instrument presents sufficient evidence of validity if it reaches optimum values in the different goodness-of-fit indices" after a confirmatory factor analysis in response to the comments raised on the validity of the measure. There are several issues with this section. To the best of my knowledge, the additional analyses included (CFA) do not provide evidence on the validity of the measure for assessing the presence of depressive symptomatology in the last 12 months: they just provide evidence on the structural/factorial validity of the measure (which was limited to the 2018 sample, and not compared to any other factor solution), and the internal consistency, which are namely related to the dimensionality of the measure and not its ability to detect symptomatology. First, CFI and TLI (and CFA, in general) cannot provide "sufficient evidence of validity", because they are indices of the goodness of fit of the model to the data. At best, they can provide evidence of the factorial/structural validity of the measure. It would be important to make this distinction in the text, mainly because the problems relative to the validity of the measure to assess depressive symptomatology cannot be answered (at least not only) by providing evidence of the measure structure. Although it seems to be somehow unidimensional (at least essentially unidimensional), the measure could be (essentially) "unidimensionally" assessing a different construct than the one intended; with no other evidences of validity, the question of whether this measure is valid for assessing depressive symptomatology remains unanswered. Second, in the main text it is not specified that these analyses were performed with the 2018 sample (this is only specified in the corresponding supplementary material). Was there any specific reason for selecting this sample over the rest of them? Evidence of factorial validity and internal consistency from the rest of the years would be of interest. In any case, at this point, it is possible that the authors are not able to obtain those additional evidences of validity, but it would be important to point at this issue in the limitations section.

DISCUSSION:

Page 37, line 51: the authors removed most references to the "five-year trends" in the text, but there is a remaining reference here

	TABLE 1: - The authors mention that they "looked for a reason for the possible variation in the wealth index". I still find confusing how these percentages can be so different considering that they come from a quintile categorisation. Could they provide the raw data on the wealth index to see how it is distributed across time points?
--	---

VERSION 2 – AUTHOR RESPONSE

Reviewer(s)' Comments to Author:

Reviewer: 1

2. The authors have done an excellent job responding to the comments and improved the manuscript greatly. Well done! I have no further comments.

Reply: Thanks

Reviewer: 2

3. The article has been improved a lot. I have no further comments.

Reply: Thanks

Reviewer: 3

GENERAL COMMENTS:

4. The article could still benefit from an additional revision of the English

Reply: The manuscript was reviewed by an external researcher, who is a native English speaker. The changes were marked in red throughout the manuscript.

ABSTRACT:

5. The "Conclusions" in the abstract merely contain results. There is no interpretation of them and/or their implications.

Reply: The conclusion in the abstract section was modified:

"No changes in trends of rates of depressive symptoms or in the proportion of depressive cases treated were observed. This suggests the need to reduce the treatment gap considering social determinants associated with inequality in access to adequate therapy."

METHODS:

6. Page 34, line 22: the commas should go after the quotation marks

Reply: This line was modified:

"There were three response options ("yes", "no", or "I do not remember")."

7. Page 32, line 31: "and the quintiles of wealth". Is this a typo? Otherwise, what are these "quintiles of wealth" that, according to this paragraph, are used in the index that will be recoded into quintiles

later on?

Reply: We agree with the reviewer, there was a typo. It was modified by:

“This index was calculated by using the availability of goods and services, the housing characteristics that the participants reported having [17]. The index was built for each year and categorized into quintiles.”

8. Statistical methods: in general, this section would benefit from a clearer presentation, starting with the main analyses, clearly outlined, then the sub-analyses, and finally the statistical software used to perform them.

Reply: The statistical methods section was modified:

“Main analysis

Firstly, the number and proportion of participants excluded from analyses were recorded (see Supplement 1). Secondly, a descriptive analysis of the participants was carried out for each year of the ENDES. Thirdly, the age-standardized prevalence of depressive symptoms was estimated using the World Health Organization (WHO) population as the reference population [24]. The age-standardized prevalence was estimated since it enabled us to compare our results with studies conducted in other countries. A 95% confidence interval (95% CI) was calculated for the prevalence at both regional and national levels. Then, an analysis was conducted on subgroup participants reporting depressive symptoms over the past year in order to determine the proportion of those who reported receiving treatment (i.e. sex, area, age groups). Finally, the trend over time was age-standardized and then evaluated using the score test for trend; for this, the year 2014 was used as the reference category. The trend test compares the odds of cases in one year with the odds of cases in the next year. This test assumes that the trend is linear and can be used in STATA with the "tabodds" command [25].

Sub-analysis

Four sub-analyses were conducted to complement the main results. In order to evaluate the measurement properties of the modified version of the PHQ-9 that collects information on depressive symptoms in the last year, a sub-analysis was conducted (see Supplement 2). A confirmatory factor analysis was performed to evaluate the validity of the modified version of the PHQ-9, considering the ordinal nature of the items and using the estimator of weighted least squares means and variance adjusted [26]. These analyses evaluate whether the instrument fits the one-dimensional model proposed by the PHQ-9, it is determined that the one-dimensional model is adequate, when optimal values are reached in the different goodness-of-fit indices. The Comparative Fit Index (CFI) and the Tucker-Lewis Index (TLI) were used; these indices must be greater than 0.90 in order to be considered of an adequate level [27]. The Root Mean Square Error of Approximation (RMSEA) with a confidence interval of 90% and the Standardized Root Mean Square Residual (SRMR) were also used, both indices considered fair values as those lower than 0.08 [27]. On the other hand, reliability was evaluated by the internal consistency coefficient of alpha and omega. Both coefficients consider that adequate levels of reliability are reached if they score higher than 0.80 [28].

An additional post hoc analysis was performed, which requires participants with depressive symptoms to meet the DSM-5 criteria. This analysis aimed to identify whether using the clinical criteria proposed by DSM-5 can alter the results (higher or lower) or be equivalent to our main results, which use a score of ≥ 15 in the PHQ-9. For this sub-analysis participants must have feelings of sadness or anhedonia (have a score of two or more on items 1 and 2 of the PHQ-9, i.e., “more than half the days” and “nearly every day”) and at least five of the other seven indicators (have a score of at least one on five of the other seven items of the PHQ-9).

In addition, a sub-analysis was performed to compare sociodemographic characteristics in people receiving treatment for depressive symptoms in the last year. The Pearson χ^2 test was used to make these comparisons. The generalized linear model assumed a Poisson distribution (crude and adjusted models). Assuming a Poisson distribution, link log, and robust variance were used as suggested in the literature [29]. Prevalence ratios (PR) and 95% CI were reported.

Finally, the age-standardized prevalence of depressive symptoms in the last two weeks and in the last year, and the proportion of depressive cases treated for each region of Peru were evaluated (Supplement 3).

Software used

All statistical analyses were performed using Stata 13 (StataCorp, College Station, TX, USA). The graphics were elaborated using the ggplot libraries in R (version 3.5.1) and QGIS v2.18. All the analyses performed considered the design by a complex sampling of the ENDES, and the analyses performed were adjusted based on the weight factor provided by each ENDES assessment year. Adjustments were made with the STATA command "svy" for all analyses, except for the factor analysis (sub-analysis) where the "lavaan" and "lavaan.survey" package was used in R."

9. In the same line, the main analysis should correspond, according to the aims of the study, to the trend analysis. The authors have changed the analytical approach for these main analyses, but the new strategy to analyse the trends is not clear.

Reply: The process involved in trend analysis was clarified:

"Finally, the trend over time was age-standardized and then evaluated using the score test for trend; for this, the year 2014 was used as the reference category. The trend test compares the odds of cases in one year with the odds of cases in the next year. This test assumes that the trend is linear and can be used in STATA with the "tabodds" command [25]."

10. The Supplement 3 is included in the text before the Supplement 2. This should be changed to reflect the order of appearance.

Reply: The order of appearance of the supplements was changed. In the Statistical Methods section and the Results section, the order of appearance of the supplements was: Supplement 1, 2, and 3 (in this order). This can be seen in the response to comment 8.

11. The authors state that "It was considered that the instrument presents sufficient evidence of validity if it reaches optimum values in the different goodness-of-fit indices" after a confirmatory factor analysis in response to the comments raised on the validity of the measure.

Reply: We agree with the reviewer's point of view. The statement was modified by:

"These analyses evaluate whether the instrument fits the one-dimensional model proposed by the PHQ-9 as the one-dimensional model seems to be adequate [20], when optimal values are reached in the different goodness-of-fit indices."

12. There are several issues with this section. To the best of my knowledge, the additional analyses included (CFA) do not provide evidence on the validity of the measure for assessing the presence of depressive symptomatology in the last 12 months: they just provide evidence on the structural/factorial validity of the measure (which was limited to the 2018 sample, and not compared to any other factor solution), and the internal consistency, which are namely related to the dimensionality of the measure and not its ability to detect symptomatology.

Reply: A confirmatory factor analysis was performed for each year of measurement, not only for 2018 (see supplement 2):

"The measurement properties of the modified version of the PHQ-9 used to assess depressive symptoms during the last year were evaluated before the primary analyses. The modified version of the PHQ-9 was identified as having evidence of validity by confirmatory factor analysis (CFI>0.90; RMSEA<0.05) and evidence of reliability by internal consistency (ω and α >0.85). More information on the factorial analysis of the modified version of the PHQ-9 for year is presented in supplement 2."

While no other possible factorial models were evaluated (e.g., two-factor models), there is evidence that the one-dimensional model is adequate. Further analysis is beyond the scope of this study.

We agree that the factorial analysis presented, although it presents evidence of validity by internal structure, may not be sufficient. So, it was added in limitations:

“Fifth, the modified version of the PHQ-9 (in the last year) may have introduced measurement bias. Although evidence of reliability and validity was presented in this study by internal structure, further studies are required to obtain other evidence of validity (i.e. relationship to other variables, invariance, sensitivity/specificity).”

12. First, CFI and TLI (and CFA, in general) cannot provide “sufficient evidence of validity”, because they are indices of the goodness of fit of the model to the data. At best, they can provide evidence of the factorial/structural validity of the measure. It would be important to make this distinction in the text, mainly because the problems relative to the validity of the measure to assess depressive symptomatology cannot be answered (at least not only) by providing evidence of the measure structure. Although it seems to be somehow unidimensional (at least essentially unidimensional), the measure could be (essentially) “unidimensionally” assessing a different construct than the one intended; with no other evidences of validity, the question of whether this measure is valid for assessing depressive symptomatology remains unanswered.

Reply:

We believe that this is a limitation that should be made clear in the text. Within the limitations, the importance of further studies on measurement validity is noted:

“Fifth, the modified version of the PHQ-9 (in the last year) may have introduced measurement bias. Although evidence of reliability and validity was presented in this study by internal structure, further studies are required to obtain other evidence of validity (i.e. relationship to other variables, invariance, sensitivity/specificity). In particular, studies on the measurement validity of the modified version of the PHQ-9 are required.”

The text also specified that:

“These results support that the modified version of the PHQ-9 presents evidence of structural validity, supporting the one-dimensional model.”

13. Second, in the main text it is not specified that these analyses were performed with the 2018 sample (this is only specified in the corresponding supplementary material). Was there any specific reason for selecting this sample over the rest of them? Evidence of factorial validity and internal consistency from the rest of the years would be of interest. In any case, at this point, it is possible that the authors are not able to obtain those additional evidences of validity, but it would be important to point at this issue in the limitations section.

Reply: A new analysis was conducted for each year:

“The measurement properties of the modified version of the PHQ-9 used to assess depressive symptoms during the last year were evaluated before the primary analyses. The modified version of the PHQ-9 was identified as having evidence of validity by confirmatory factor analysis (CFI>0.90; RMSEA<0.05) and evidence of reliability by internal consistency (ω and α >0.85). More information on the factorial analysis of the modified version of the PHQ-9 for year is presented in supplement 2.”

DISCUSSION:

14. Page 37, line 51: the authors removed most references to the “five-year trends” in the text, but there is a remaining reference here

Reply: The sentence was changed to:

“Although our work only analyzed between 2014 and 2018, it is perhaps the first to report estimates across consecutive years using nationally representative surveys.”

TABLE 1:

15. The authors mention that they "looked for a reason for the possible variation in the wealth index". I still find confusing how these percentages can be so different considering that they come from a quintile categorization. Could they provide the raw data on the wealth index to see how it is distributed across time points?

Reply: We consider that the variations are not so great. We must consider that the procedures we carried out to obtain these analyses were: A) Establish the wealth quintiles. B) Apply the inclusion criteria (cases eliminated from supplement 1). C) Obtain the socio-demographic characteristics (cases in Table 1).

In the case of eliminated participants (supplement 1), the percentage of eliminated cases per wealth quintile is very similar in our opinion. For example, in 2014 the percentages of participants eliminated per quintile range from 17.8% to 23.9%. While there may be loss mechanisms that have been evaluated, we believe that this is beyond the scope of the study. Therefore, it was added in limitation: “Fourth, possible loss mechanisms in the missing data were not evaluated.”

As for the percentage by wealth quintile in Table 1, it should be considered that this variable was weighted by the expansion factor of the complex sampling. So, it is understandable that each wealth category (quintile) will not be exactly 20%. We consider that the percentages are very similar. For example, in 2014 the values vary between 18.7% and 21.2%.

VERSION 3 – REVIEW

REVIEWER	Darío Moreno-Agostino King's College London
REVIEW RETURNED	18-May-2020
GENERAL COMMENTS	The authors have addressed all the points raised. The manuscript has substantially improved. I have no further comments.